# Understanding the Connection between Low-Dimensional Representation and Generalization via Interpolation

## Abstract

In recent years, numerous studies have demonstrated the close connection between neural networks' generalization performance and their ability to learn low-dimensional representations of data. However, the theoretical foundation linking low-dimensional representations to generalization remains underexplored. In this work, we propose a theoretical framework to analyze this relationship from the perspective of interpolation and convex combinations. We argue that lower-dimensional representations increase the likelihood of new samples being expressed as convex combinations of the training set, thereby enhancing interpolation probability. We derive a generalization error upper bound under the interpolation regime, which becomes tighter as the dimensionality of the representation decreases. Furthermore, we investigate how the structure of the manifold affects interpolation probability by examining the volume of the convex hull formed by the manifold. Our theoretical and experimental results show that larger convex hull volumes are associated with higher interpolation probabilities. Additionally, we explore the impact of training data volume on interpolation, finding a significant power-law relationship between increased data volume, convex hull volume and interpolation probability. Overall, this study highlights the critical role of low-dimensional representations in improving the generalization performance of neural networks, supported by both theoretical insights and experimental evidence.

## 1 Introduction

Neural networks have achieved remarkable success across various tasks, often outperforming humans in domains like natural language processing and computer vision (Devlin et al., 2018; Brown et al., 2020; Raffel et al., 2020; Yang et al., 2019; Liu et al., 2019; He et al., 2022). However, understanding the generalization capabilities of these models remains an open problem. Current theories fail to explain how large-scale data and models contribute to this generalization, leading to an overemphasis on increasing data and model size without clear theoretical backing.

Recent studies have revealed several phenomena relevant to generalization in neural networks. For instance, models with flatter minima often generalize better (Hochreiter & Schmidhuber, 1997; Mulayoff & Michaeli, 2020; Baldassi et al., 2021), and networks tend to perform better in interpolation tasks, where test samples lie within the convex hull of the training data (Barnard & Wessels, 1992; Haley & Soloway, 1992). Additionally, there is growing evidence that neural networks compress input data into low-dimensional representations, which appears to improve generalization (Ansuini et al., 2019; Recanatesi et al., 2019).

Despite these insights, there is still no unified framework to explain why low-dimensional representations correlate with better generalization. Furthermore, the connection between interpolation probability and low-dimensional representation has not been thoroughly ex-

plored. This study aims to bridge this gap by proposing a theoretical framework that links low-dimensional representations, interpolation probability and generalization error.

We hypothesize that lower-dimensional representations increase the likelihood of new samples being represented as convex combinations of the training set, resulting in a higher interpolation probability. This, in turn, tightens the generalization error bound. Through both theoretical derivation and empirical validation, we demonstrate that neural networks improve generalization by learning low-dimensional representation manifolds.

The structure of this paper is as follows: Section 2 reviews related work on neural network generalization. Section 3 introduces the relevant background knowledge for this research. Section 4 experimentally analyzes how the embedding dimension, manifold structure, and interpolation probability change during the training process of neural networks. Section 5 proves that in the interpolation regime, there exists an upper bound on the generalization error of neural networks, which decreases as the data dimension decreases. Section 6 theoretically discusses the relationship between dimension and interpolation probability. Section 7 theoretically examines the relationship between manifold structure (primarily the volume of convex hull formed by the manifold) and interpolation probability. Finally, Section 8 explores the impact of data volume on interpolation probability from both theoretical and experimental perspectives.

Our key contributions are as follows:

- We empirically show that during training, neural networks reduce the dimensionality of their representations while increasing interpolation probability, leading to improved generalization performance.
- We provide a theoretical framework that demonstrates how low-dimensional representations increase interpolation probability, which in turn leads to tighter generalization error bounds in the interpolation regime.
- We explore how the structure of the representation manifold, particularly the volume of the convex hull, influences interpolation probability, and we show that increased training data expands this volume, enhancing generalization.

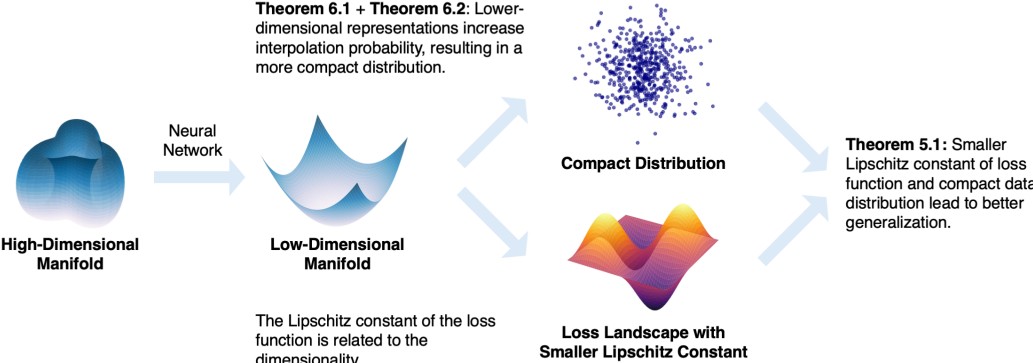

Figure 1: **Theoretical Framework.** By learning low-dimensional representations, neural networks can achieve a more compact embedding distribution, which in turn results in a loss landscape with lower Lipschitz constants around the learned parameters. In the interpolation regime, this compactness of the embeddings, combined with the reduced Lipschitz constant, contributes to a tighter upper bound on the generalization error, enhancing the model's ability to generalize effectively.

## 2 RELATED WORKS

**Interpolation and Extrapolation in Neural Networks**: For neural networks, interpolation refers to test samples falling within the convex hull of training samples, while extrapolation refers to test samples lying outside this convex hull. Generally, neural networks

tend to exhibit better generalization performance when performing interpolation (Barnard & Wessels, 1992; Haley & Soloway, 1992). Recent work highlights that interpolation probability is related to the amount of data constituting the convex hull and the dimensionality of the data (Bárány & Füredi, 1988; Balestriero et al., 2021). Given that the data processed by models are typically high-dimensional and often insufficient in quantity to meet theoretical requirements, interpolation probabilities are theoretically quite low (Balestriero et al., 2021). Thus, it is believed that neural networks are likely always performing extrapolation.

**Low-dimensional Representation of Neural Network**: Numerous studies indicate that neural networks compress data, and this compression capability is closely linked to their generalization performance (Yu et al., 2024; Dai et al., 2023; Chen et al., 2022; Chan et al., 2022). The intrinsic dimension can quantify the dimensionality of the manifold where a discrete set of points resides, serving as a measure of manifold complexity. By comparing the differences in intrinsic dimension between raw data and its representations, we can characterize the extent of compression achieved by the neural network. Research shows that as data progresses through each layer from raw input to the embedding manifold, the intrinsic dimension of the embedding manifold consistently decreases, reflecting ongoing compression of the data (Ansuini et al., 2019; Recanatesi et al., 2019). Furthermore, lower-dimensional representations often correlate with better generalization performance (Ansuini et al., 2019). While there is widespread recognition of the importance of low-dimensional representations for enhancing generalization, a theoretical explanation for why low-dimensional representations tend to perform better has yet to be established.

## 3 PRELIMINARIES AND TECHNICAL BACKGROUND

### 3.1 CONVEX HULL AND INTERPOLATION

**Definition 1.** *Convex Hull: Given a set of points $X = \{x_1, x_2, \ldots, x_n\} \subset \mathbb{R}^d$, the **convex hull** of $X$ is defined as:*

$$Conv(X) = \left\{ \sum_{i=1}^{n} \lambda_i x_i \,\middle|\, \lambda_i \geq 0, \sum_{i=1}^{n} \lambda_i = 1 \right\}. \tag{1}$$

This represents the smallest convex set containing $X$.

**Definition 2.** *Volume of a Convex Hull: Let $S = \{x_1, x_2, \ldots, x_m\} \subset \mathbb{R}^n$ be a finite set of points in $n$-dimensional Euclidean space. The volume of the convex hull, $\lambda_n(Conv(S))$, can be computed using Lebesgue integration as:*

$$\lambda_n(Conv(S)) = \int_{\mathbb{R}^n} \chi_{Conv(S)}(x) \, d\lambda_n(x), \tag{2}$$

*where $\chi_{Conv(S)}(x)$ is the characteristic function of the convex hull $Conv(S)$, defined as:*

$$\chi_{Conv(S)}(x) = \begin{cases} 1, & \text{if } x \in Conv(S), \\ 0, & \text{otherwise.} \end{cases} \tag{3}$$

*Thus, the volume $\lambda_n(Conv(S))$ is the $n$-dimensional Lebesgue measure of the set $Conv(S)$, which generalizes the concept of the volume of a convex polytope.*

**Definition 3.** *Interpolation Probability: Let $X$ be a $d$-dimensional random vector and $X_1, X_2, \ldots$ be independent copies of $X$. For each $\theta \in \mathcal{R}^d$ and positive integer $n$, define*

$$p_{n,X}(\theta) := \mathcal{R}(\theta \in Conv\{X_1, \ldots, X_n\}), \tag{4}$$

*where $Conv\, A := \{\sum_{i=1}^{m} \lambda_i x_i | m \geq 1, x_i \in A, \lambda_i \geq 0, \sum_{i=1}^{m} \lambda_i = 1\}$ denotes the convex hull of a set $A \subset \mathcal{R}^d$.*

In this work, we will focus on the relationship between the convex hull volume and interpolation probability. However, the dimensionality also affects the convex hull volume, so we need to better understand the interplay between these three factors. Using the high-dimensional sphere as an example, the following theorem illustrates how the volume of a geometric body changes as the dimension increases.

**Theorem 3.1** (Volume of a $d$-dimensional unit sphere). *Let $S^d$ denote the unit sphere in $\mathbb{R}^{d+1}$, that is, the set of points in $\mathbb{R}^{d+1}$ that are at a distance of 1 from the origin. The volume $V_d$ of the d-dimensional unit sphere is given by:*

$$V_d = \frac{\pi^{d/2}}{\Gamma\left(\frac{d}{2}+1\right)} \tag{5}$$

*where $\Gamma(x)$ is the Gamma function, which generalizes the factorial function, such that $\Gamma(n) = (n-1)!$ for positive integers $n$.*

***Asymptotic behavior:***

- *For small dimensions, $V_d$ increases with $d$, reaching a maximum at a certain dimension (approximately around $d = 5$ to $d = 9$).*

- *For large dimensions, $V_d$ decreases rapidly and approaches zero as $d \to \infty$.*

### 3.2 INTRINSIC DIMENSION AND AMBIENT DIMENSION

Let $\mathcal{P} \subset R^N$ represent a set of sample points. We assume that these points lie on a low-dimensional manifold $\mathcal{M} \subset R^N$, where $N$ is the ambient dimension of the space. The ambient dimension $dim(R^N) = N$ refers to the dimension of the surrounding space, while the intrinsic dimension $dim(\mathcal{M}) = d \ll N$ refers to the dimension of the manifold on which the data lies. In essence, the intrinsic dimension quantifies the complexity of the underlying structure of the data.

### 3.3 ESTIMATION OF THE INTRINSIC DIMENSION

To estimate the intrinsic dimension of a manifold, we employ the Maximum Likelihood Estimation (MLE) method proposed by Levina et al. (Levina & Bickel, 2004). This technique relies on the distances between neighboring points in the dataset to compute the manifold's intrinsic dimension.

The intrinsic dimension $\hat{m}_k(x)$ at a point $x$ can be estimated as follows:

$$\hat{m}_k(x) = \left[\frac{1}{k-1}\sum_{j=1}^{k-1} log\frac{T_k(x)}{T_j(x)}\right]^{-1}, \tag{6}$$

where $T_j(x)$ denotes the Euclidean distance from point $x$ to its $j^{th}$ nearest neighbor. By averaging these local estimates across all samples, we obtain a global estimate for the intrinsic dimension:

$$\bar{m}_k = \frac{1}{n}\sum_{i=1}^{n} \hat{m}_k(x_i), \tag{7}$$

The parameter $k$ controls the number of neighbors considered when estimating the dimension. A smaller $k$ focuses on a more local perspective, while a larger $k$ captures a more global view of the manifold. By varying $k$, we can derive a more comprehensive understanding of the manifold's intrinsic dimension.

## 4 EXPERIMENT RESULTS OF DIMENSIONAL AND STRUCTURAL ANALYSIS

In this experiment, we evaluate a 5-layer Multi-Layer Perceptron (MLP) model trained on the MNIST dataset and report findings on three critical metrics: the intrinsic dimension of embeddings, convex hull volume, and interpolation probability. The embeddings used in this analysis are taken from the output of the third layer's linear transformation. The convex hull volume provides a geometric characterization of the manifold structure, encapsulating

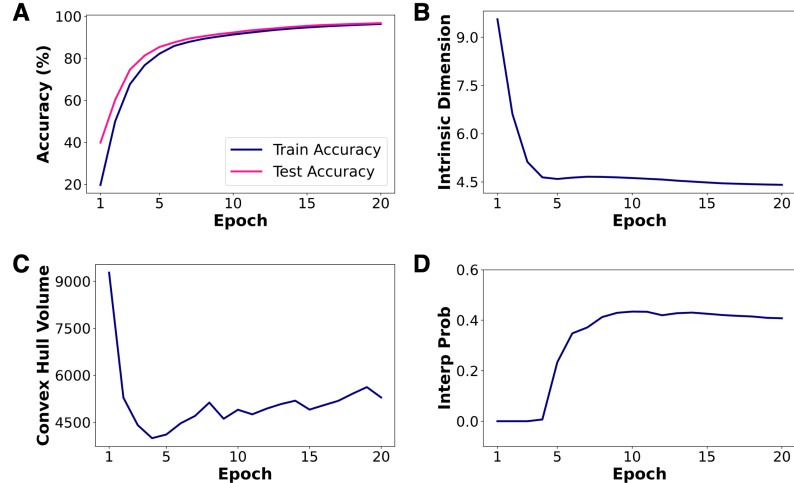

Figure 2: **Learning dynamics of Neural Network on MNIST.** (A) Train and test accuracy curves show rapid convergence to high accuracy. (B) Intrinsic dimension decreases sharply in the early epochs, stabilizing as training progresses. (C) Convex hull volume initially drops, followed by a gradual increase, indicating changes in the embedding structure. (D) Interpolation probability remains low early in training but begins to rise gradually once the intrinsic dimension decreases to a certain level.

the distributional shape of the data. Additional results for various model architectures on other datasets are provided in the appendix.

As shown in Figure 2, intrinsic dimension decreases rapidly in the early stages of training and stabilizes midway, suggesting that the model quickly captures lower-dimensional representations before entering a phase of fine-tuning. For the convex hull volume, it initially decreases rapidly and then rises again. The initial decrease is primarily due to the reduction in intrinsic dimension, while the subsequent increase occurs when the embedding dimension remains relatively stable, indicating that the structure of the embedding manifold continues to adjust during this phase. In terms of interpolation probability, it starts at zero due to the initially high intrinsic dimension. Later, as both the intrinsic dimension and the manifold structure evolve, the interpolation probability begins to gradually increase.

The above results demonstrate the complex changes in the embedding manifold during the learning process of neural networks, highlighting the close relationship between intrinsic dimension, convex hull volume, interpolation probability and the model's generalization performance. Next, we will further explore, from a theoretical perspective, how these factors influence generalization performance.

## 5 Existence of Generalization Error Bound in the Interpolation Regime and the Impact of Dimension

In this section, we primarily prove that in the interpolation regime, an upper bound on the generalization error exists, and this upper bound is related to the dimension of the input data. The smaller the dimension, the lower the upper bound on the error.

**Theorem 5.1.** *Let $\ell(y, x, \theta)$ be a loss function that is Lipschitz continuous with respect to both $x \in \mathbb{R}^d$ and $y \in \mathbb{R}^k$, with Lipschitz constant $L$. Assume that the input data $x$ and output data $y$ are bounded such that $\|x - x'\| \leq D_x$ and $\|y - y'\| \leq D_y$ for all $x, x'$ and $y, y'$. Let $\hat{L}(\theta, D)$ be the empirical loss over a dataset $D = \{(x_i, y_i)\}_{i=1}^n$, and let $L(\theta)$ be the expected loss over the data distribution $v$. Then, for any $\epsilon > 0$, the following bound holds:*

$$P\left(\left|\hat{L}(\theta, D) - L(\theta)\right| \geq \epsilon\right) \leq 2 \exp\left(-\frac{2n\epsilon^2}{L^2(D_x + D_y)^2}\right). \tag{8}$$

*Furthermore, if the Lipschitz constant $L$ and the data diameters $D_x$ and $D_y$ scale with the dimension $d$ as $L = C_L\sqrt{d}$ and $D_x = C_x\sqrt{d}$, while $D_y$ is constant, then the bound becomes:*

$$P\left(\left|\hat{L}(\theta, D) - L(\theta)\right| \geq \epsilon\right) \leq 2\exp\left(-\frac{2n\epsilon^2}{C^2 d^2}\right), \tag{9}$$

*where $C = C_L(C_x + C_y/\sqrt{d})$ and for large $d$, $C \approx C_L C_x$. This shows that the generalization error bound becomes tighter as the dimension $d$ decrease.*

Theorem 5.1 emphasizes the impact of the Lipschitz constant of the loss function and the diameter of the data distribution on generalization performance. A smaller Lipschitz constant for the loss function indicates that changes in model parameters will not lead to significant fluctuations in the loss value, which aligns with the definition of flat minima. Therefore, this theorem explains why flat minima often exhibit better generalization performance.

In terms of data distribution, under the interpolation regime, the distribution is bounded, but it is worth further exploring the diameter of the data distribution. The diameter reflects the maximum distance between samples in the space, and a smaller diameter indicates a more compact data distribution. Compactness can be understood as the degree of concentration in the data distribution, allowing the model to capture data features more effectively. However, as the diameter of the data distribution decreases, the model's sensitivity to input perturbations may increase, leading to a larger Lipschitz constant for the loss function. This is because a compact embedding distribution can amplify the effects of small changes, causing the model to respond more sharply to these perturbations, which in turn affects the smoothness of the loss function and the model's generalization performance.

Additionally, this theory highlights the impact of data dimensionality on generalization performance. However, since the dimensionality of the original data is usually fixed, the theorem applies more to the model's representation of the original data. This theorem can explain why models with lower-dimensional embeddings tend to generalize better.

## 6 Impact of Manifold Dimension on Interpolation Probability

The analysis of Theorem 5.1 is based on the assumption of interpolation. Therefore, we aim to identify the factors that influence interpolation probability. In this section, we will analyze the relationship between dimension and interpolation probability

**Theorem 6.1** ((Bárány & Füredi, 1988)). *Given a $d$-dimensional dataset $X \triangleq x_1, ..., x_N$ with i.i.d. samples uniformly drawn from a hyperball, the probability that a new sample $x$ is in the interpolation regime exhibits the following asymptotic behavior:*

$$lim_{d\to\infty} p(x \in Conv(X)) = \begin{cases} 1 \Leftrightarrow N > d^{-1}2^{d/2} \\ 0 \Leftrightarrow N < d^{-1}2^{d/2} \end{cases} \tag{10}$$

**Theorem 6.2** ((Kabluchko & Zaporozhets, 2020)). *Let $X$ consist of $N$ i.i.d. $d$-dimensional samples from $\mathbb{N}(0, I_d)$ with $N \geq d+1$, then for every $\sigma \geq 0$ the probability that a new sample $x \sim \mathbb{N}(0, \sigma^2 Id)$ is in extrapolation regime is given by*

$$p(x \notin Conv(X)) = 2(b_{N,d-1}(\sigma^2) + b_{N,d-3}(\sigma^2) + ...) \tag{11}$$

*with*

$$b_{n,k}(\sigma^2) = \binom{n}{k}g_k(-\frac{\sigma^2}{1+k\sigma^2})g_{n-k}(\frac{\sigma^2}{1+k\sigma^2}), \ g_n(r) = \frac{1}{\sqrt{2\pi}}\int_{-\infty}^{\infty}\Phi^n(\sqrt{r}x)e^{-x^2/2}dx \tag{12}$$

*where $\sqrt{r} = i\sqrt{-r}$ if $r < 0$ and $b_{N,k} = 0$ for $k \notin \{0, 1, ..., N\}$.*

Theorem 6.1 indicates that as dimensionality increases, the convex hull struggles to cover the entire data space, causing a significant drop in interpolation probability. In high-dimensional spaces, maintaining a high interpolation probability requires an exponential increase in data

size. In contrast, in low-dimensional spaces, data points are denser, making it easier for the convex hull to cover new samples, resulting in a higher interpolation probability.

Theorem 6.2 quantitatively describes the probability of extrapolation in high-dimensional spaces. As dimensionality increases, the likelihood of extrapolation rises, and interpolation probability decreases.

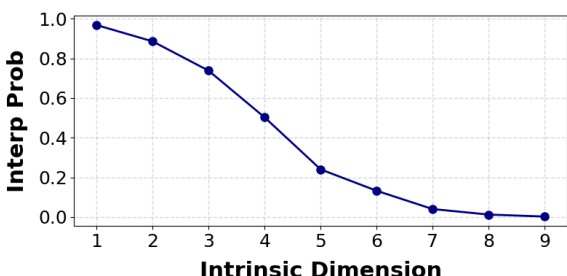

Figure 3: **Interpolation Probability Estimation of Data on Hypersphere of Different Dimensions**. As the dimension of the hypersphere increases, with a constant amount of data forming the convex hull, the probability that a new sample falls within the convex hull continuously decreases.

We also verified the relationship between interpolation probability and dimension by generating data distributed on hypersphere of different dimensions. The ambient dimension of the data was fixed at 10, with an intrinsic dimension (dimension of hypersphere) ranging from 1 to 9. For each intrinsic dimension, we generated 1000 samples, randomly selected 100 samples to construct the convex hull, and used the remaining 900 samples to calculate the interpolation probability. As shown in Figure 3, the interpolation probability significantly decreases as the intrinsic dimension of the samples increases.

## 7 Impact of Manifold Structure on Interpolation Probability

In this section, we will explore the relationship between manifold structure and interpolation probability, focusing on the volume of convex hull formed by the manifold.

**Proposition 1** (Relationship between Convex Hull Volume and Interpolation Probability)**.** *Let $M \subseteq \mathbb{R}^n$ be a compact convex set, and let $\mu$ be a probability measure on $\mathbb{R}^n$ with density function $f(x)$ satisfying $f(x) \geq c > 0$ for all $x \in M$ and $f(x) = 0$ for $x \notin M$. Suppose $N \geq n + 1$ points $x_1, x_2, \ldots, x_N$ are independently sampled from $M$ according to $\mu$, and let $C = \mathrm{Conv}(x_1, x_2, \ldots, x_N)$ be the convex hull of these points. Then, the probability that a newly sampled point from $M$ falls inside $C$ is given by*

$$\mu(C) = \int_C f(x) \, d\lambda(x) \geq c \cdot \mathrm{Vol}(C). \tag{13}$$

Thus, the probability $\mu(C)$ that a new point falls inside $C$ increases as the volume $\mathrm{Vol}(C)$ increases.

We further validate the relationship between convex hull volume and interpolation probability through experiments. We repeatedly generated three types of two-dimensional random datasets, each distributed across different manifold shapes. For each dataset, we computed the convex hull volume and the corresponding interpolation probability. As illustrated in Figure 4, when the manifold shape is held constant, a larger convex hull volume is associated with a higher interpolation probability. Interestingly, we observed notable differences in interpolation probability even when the convex hull volumes were similar. For example, the interpolation probabilities for the triangle and the circle were comparable, yet the convex hull volume of the triangle was significantly smaller than that of the circle. This suggests that while convex hull volume provides some insight into changes in manifold structure and correlates with interpolation probability, future work should aim for a more comprehensive analysis that considers multiple aspects of manifold geometry.

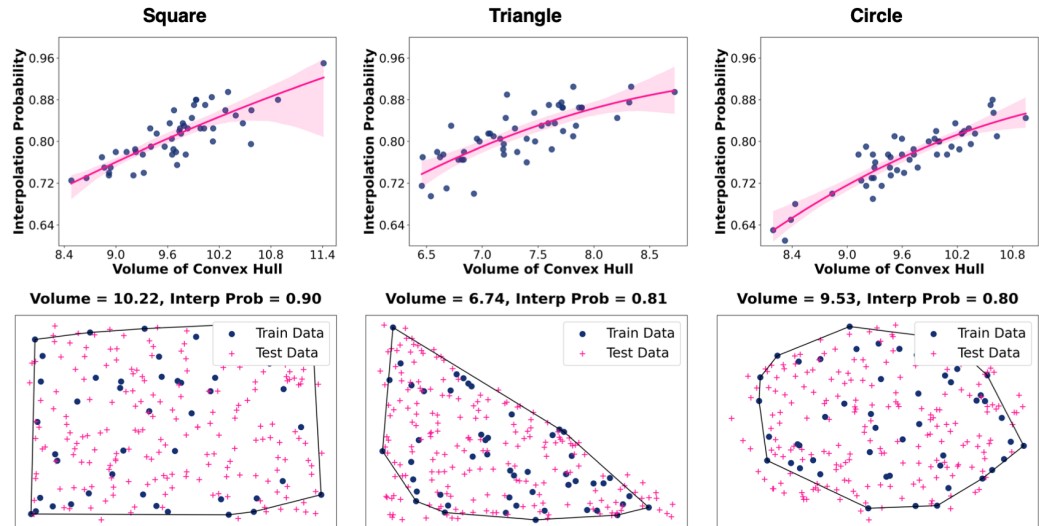

Figure 4: **Influence on Interplolation Probability of Manifold Volume.** The top row shows the relationship between convex hull volume and interpolation probability for three different shapes: square, triangle, and circle. A positive correlation is observed, where larger volumes lead to higher interpolation probabilities. The bottom row provides specific examples corresponding to each shape, with convex hulls formed by training data (blue dots) and test data (pink crosses).

## 8 IMPACT OF DATA VOLUME ON INTERPOLATION PROBABILITY

In the previous analysis, we primarily focused on the impact of the manifold's dimensionality and structure on interpolation probability under a fixed data regime. However, it is well understood that, with the same model architecture, increasing the size of the training data often leads to better performance. Therefore, we aim to theoretically explore how larger datasets affect interpolation probability.

**Proposition 2** ((Hayakawa et al., 2023)). *For an arbitrary d-dimensional random vector X with $\mathcal{E}[x] = 0$ and $\mathcal{P}(x \neq 0) > 0$, we have*

$$0 < P_{d+1,X} < P_{d+2,X} < ... < P_{n,X} < ... \rightarrow 1. \tag{14}$$

*The conclusion still holds if we only assume $p_{n,X} > 0$ for some n instead of $\mathcal{E}[X] = 0$*

This proposition highlights that as the number of data points $n$ increases, the interpolation probability $P_{n,X}$ rises monotonically and approaches 1. This means that with a sufficiently large dataset, new samples are almost always interpolated within the convex hull of the existing data. Importantly, the result holds even if we only assume a positive interpolation probability for some finite $n$, making it applicable to a wide range of distributions.

We further validated this proposition through experiments. Specifically, we generated data of the same dimensionality and constructed their convex hulls, calculating the corresponding convex hull volume and interpolation probability for different data sizes. Our experimental results in Figure 5 show that as the data size increases, both the interpolation probability and convex hull volume grow steadily, and the rate of increase follows an approximate power law with respect to data size.

This behavior is consistent with various scaling laws observed in deep learning, such as the relationship between model performance and the increase in data size and model parameters (Kaplan et al., 2020; Gordon et al., 2021). In these scaling laws, although larger datasets can improve model generalization, the benefits exhibit diminishing returns. This aligns with our observation of the convex hull volume expansion: as data size grows, the expansion of the convex hull volume saturates, and the gains in interpolation probability gradually

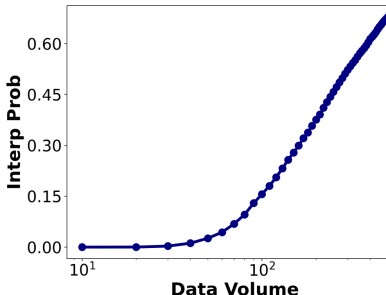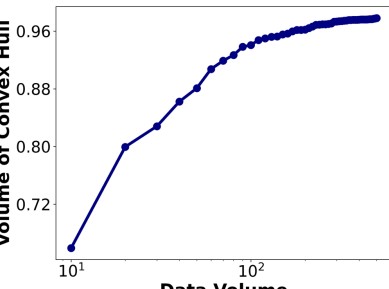

Figure 5: **The effect of data volume on interpolation probability and convex hull volume.** The left plot shows that interpolation probability increases as the data volume grows, following a scaling law, where larger datasets improve interpolation capabilities. The right plot demonstrates that the convex hull volume also scales with data volume, though at a slower rate compared to the rise in interpolation probability.

diminish. These findings provide a geometric interpretation of how larger datasets contribute to improved model generalization.

## 9 Discussion and Conclusion

This paper introduces a novel framework that connects low-dimensional representations and interpolation probabilities to the generalization capabilities of neural networks. Our approach also integrates manifold structure (convex hull volume) into a unified theoretical explanation for generalization, supported by both empirical validation and theoretical proofs.

The key contributions are: (1) A theoretical framework that links interpolation, manifold geometry and generalization, offering new insights into why low-dimensional, compact representations lead to improved generalization. (2) Empirical results demonstrating the relationship between Intrinsic dimension, convex hull volume, interpolation probability and generalization. (3) Theoretical exploration of scaling laws showing how increasing data volume affects interpolation probability.

However, the framework has certain limitations. Simplified assumptions about uniform data distribution and stable manifold structures may not fully capture the complexity of real-world datasets (Cooper & Green, 2017; Majeed, 2019), and the evolution of neural network embedding manifolds is also far more dynamic and intricate than such assumptions suggest (Kunin et al., 2020). Moreover, the focus on interpolation might overlook scenarios where extrapolation is required for robust generalization, particularly in tasks involving out-of-distribution data (Liu et al., 2021; Li et al., 2024). The experiments, primarily on MNIST and Cifar10, should also be extended to more complex datasets for broader validation.

Moreover, while convex hull volume offers valuable insights into how manifold structure influences interpolation probabilities, it assumes a relatively stable manifold structure. In practice, however, manifold structures evolve continuously throughout the learning process (Birdal et al., 2021; Magai & Ayzenberg, 2022), rendering convex hull volume an incomplete metric. To fully capture the dynamic nature of representation manifolds during neural network training, a more comprehensive structural quantification method is needed. Such a method should also be interpretable, providing insights into both how and why these structural changes contribute to improved generalization performance.

Future work should address these limitations by considering more realistic data distributions and validating the results on larger, more diverse datasets. Additionally, exploring how different neural network architectures influence manifold learning and interpolation probability could further enhance the model's generalization capabilities.

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

## A  SUPPLEMENTARY ANALYSIS OF LEARNING DYNAMICS

To verify the generalizability of the results in Section 4, we also analyzed multiple model architectures (e.g., AlexNet, VGG16) across different datasets (e.g., CIFAR10, MNIST), as shown in Figure 6. The findings generally align with the theoretical trends discussed in the main text: as training progresses, classification performance improves, the intrinsic dimension of embeddings decreases, the convex hull volume first drops and then gradually rises, and interpolation probability increases. However, different datasets and model architectures introduce some subtle variations. For instance, when training CIFAR10 with AlexNet, we observed that after an initial decrease, the intrinsic dimension begins to rise again, which may be related to AlexNet's relatively poor performance on CIFAR10 and warrants further investigation. Additionally, in the convex hull volume estimation, occasional abrupt changes were noted, suggesting that the current convex hull estimation algorithms may have limitations in high-dimensional spaces. Therefore, developing more accurate and efficient algorithms for estimating convex hull volume in high-dimensional spaces is a key area for future work.

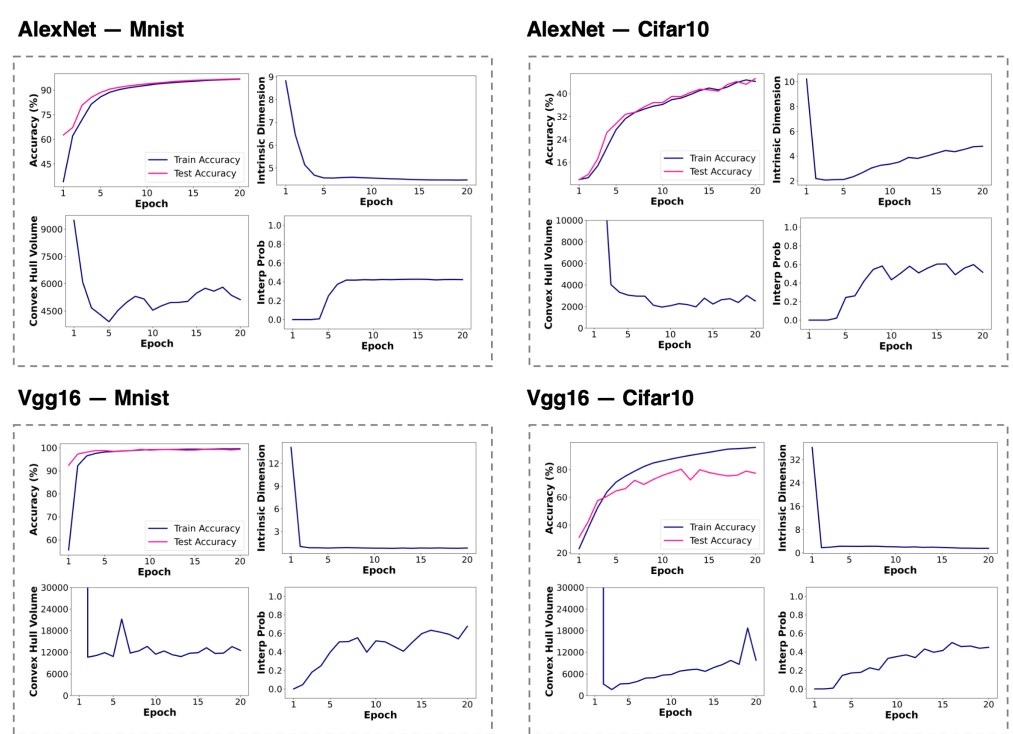

Figure 6: **Supplementary Analysis of Learning Dynamics.** We conducted training on CIFAR10 using both AlexNet and VGG16. The trends in intrinsic dimension, convex hull volume, and interpolation probability during the training process remained largely consistent across both models.

## B  PROOF OF THEOREM 5.1

**Definitions   Empirical Loss**:

$$\hat{L}(\theta, D) = \frac{1}{n} \sum_{i=1}^{n} \ell(y_i, x_i, \theta),$$

where $D = \{(x_i, y_i)\}_{i=1}^{n}$ is the dataset.

**Expected Loss**:

$$L(\theta) = \mathbb{E}_{(x,y) \sim v}[\ell(y, x, \theta)],$$

where $v$ is the data distribution.

**Objective**   Our goal is to bound the probability:

$$P\left(\left|\hat{L}(\theta, D) - L(\theta)\right| \geq \epsilon\right).$$

**Step 1: McDiarmid's Inequality**   McDiarmid's inequality states that if $X_1, X_2, \ldots, X_n$ are independent random variables taking values in a set $A$, and the function $f : A^n \to \mathbb{R}$ satisfies the bounded differences condition:

$$\sup_{x_1,\ldots,x_n,x_i'} |f(x_1, \ldots, x_i, \ldots, x_n) - f(x_1, \ldots, x_i', \ldots, x_n)| \leq c_i,$$

then for all $\epsilon > 0$:

$$P\left(f(X) - \mathbb{E}[f(X)] \geq \epsilon\right) \leq \exp\left(-\frac{2\epsilon^2}{\sum_{i=1}^n c_i^2}\right).$$

**Step 2: Bounded Differences Condition**   We need to verify the bounded differences condition for the empirical loss function $\hat{L}(\theta, D)$ when one sample $(x_i, y_i)$ is replaced by another $(x_i', y_i')$.

Define:

$$\Delta_i = \left|\hat{L}(\theta, D) - \hat{L}(\theta, D_i')\right|,$$

where $D_i'$ is the dataset $D$ with the $i$-th sample replaced by $(x_i', y_i')$.

Compute $\Delta_i$:

$$\Delta_i = \left|\frac{1}{n}\left(\ell(y_i, x_i, \theta) - \ell(y_i', x_i', \theta)\right)\right|.$$

**Step 3: Applying Lipschitz Continuity**   By the Lipschitz continuity of $\ell$, we have:

$$|\ell(y_i, x_i, \theta) - \ell(y_i', x_i', \theta)| \leq L\left(\|x_i - x_i'\| + \|y_i - y_i'\|\right).$$

Therefore,

$$\Delta_i \leq \frac{L}{n}\left(\|x_i - x_i'\| + \|y_i - y_i'\|\right).$$

Using the boundedness of the data:

$$\|x_i - x_i'\| \leq D_x, \quad \|y_i - y_i'\| \leq D_y,$$

so we have:

$$\Delta_i \leq \frac{L}{n}(D_x + D_y) = c_i.$$

**Step 4: Calculating the Sum of $c_i^2$**   Since $c_i = \frac{L}{n}(D_x + D_y)$ for all $i$, we have:

$$\sum_{i=1}^n c_i^2 = nc_i^2 = n\left(\frac{L}{n}(D_x + D_y)\right)^2 = \frac{L^2(D_x + D_y)^2}{n}.$$

**Step 5: Applying McDiarmid's Inequality**   Applying McDiarmid's inequality:

$$P\left(\hat{L}(\theta, D) - \mathbb{E}[\hat{L}(\theta, D)] \geq \epsilon\right) \leq \exp\left(-\frac{2\epsilon^2}{\sum_{i=1}^n c_i^2}\right) = \exp\left(-\frac{2n\epsilon^2}{L^2(D_x + D_y)^2}\right).$$

Similarly, for the lower tail:

$$P\left(\hat{L}(\theta, D) - \mathbb{E}[\hat{L}(\theta, D)] \leq -\epsilon\right) \leq \exp\left(-\frac{2n\epsilon^2}{L^2(D_x + D_y)^2}\right).$$

Combining both tails:

$$P\left(\left|\hat{L}(\theta, D) - \mathbb{E}[\hat{L}(\theta, D)]\right| \geq \epsilon\right) \leq 2\exp\left(-\frac{2n\epsilon^2}{L^2(D_x + D_y)^2}\right).$$

**Step 6: Connecting to Expected Loss** Since samples are independent and identically distributed (i.i.d.) from distribution $v$, we have:

$$\mathbb{E}[\hat{L}(\theta, D)] = L(\theta).$$

Therefore:

$$P\left(\left|\hat{L}(\theta, D) - L(\theta)\right| \geq \epsilon\right) \leq 2 \exp\left(-\frac{2n\epsilon^2}{L^2(D_x + D_y)^2}\right).$$

This proves the first part of the theorem.

**Step 7: Dependence on Dimension $d$** Assume the following scaling with dimension $d$:

1. **Lipschitz Constant $L$:**

$$L = C_L \sqrt{d},$$

where $C_L$ is a constant independent of $d$.

2. **Data Diameter $D_x$:**

$$D_x = C_x \sqrt{d},$$

where $C_x$ is a constant.

3. **Data Diameter $D_y$:** For simplicity, assume $D_y$ is constant (i.e., the dimension of $y$ does not grow with $d$).

**Step 8: Substituting into the Bound** Compute the denominator in the exponent:

$$L^2(D_x + D_y)^2 = (C_L \sqrt{d})^2 (C_x \sqrt{d} + D_y)^2 = C_L^2 d (C_x \sqrt{d} + D_y)^2.$$

For large $d$, $C_x \sqrt{d}$ dominates $D_y$, so:

$$C_x \sqrt{d} + D_y \approx C_x \sqrt{d}.$$

Thus,

$$L^2(D_x + D_y)^2 \approx C_L^2 d (C_x \sqrt{d})^2 = C_L^2 d (C_x^2 d) = C_L^2 C_x^2 d^2.$$

Therefore, the bound becomes:

$$P\left(\left|\hat{L}(\theta, D) - L(\theta)\right| \geq \epsilon\right) \leq 2 \exp\left(-\frac{2n\epsilon^2}{C_L^2 C_x^2 d^2}\right).$$

Let $C = C_L C_x$, so:

$$P\left(\left|\hat{L}(\theta, D) - L(\theta)\right| \geq \epsilon\right) \leq 2 \exp\left(-\frac{2n\epsilon^2}{C^2 d^2}\right).$$

This proves the second part of the theorem.

**Conclusion** The bound on the generalization error becomes tighter as the dimension $d$ decreases, specifically due to the $d^2$ term in the denominator of the exponent. This indicates that in lower-dimensional spaces, fewer samples $n$ are required to ensure that the empirical loss $\hat{L}(\theta, D)$ closely approximates the expected loss $L(\theta)$. Therefore, reducing the dimensionality of the input data can significantly improve generalization performance and reduce the risk of overfitting, highlighting the importance of low-dimensional representation for generalization.

## C    Proof of Proposition 1

The probability that a newly sampled point $x \in M$ falls inside the convex hull $C = \text{Conv}(x_1, x_2, \ldots, x_N)$ is expressed as

$$\mu(C) = \int_C f(x) \, d\lambda(x).$$

Given that $f(x) \geq c > 0$ for all $x \in C$, we derive

$$\mu(C) = \int_C f(x) \, d\lambda(x) \geq \int_C c \, d\lambda(x) = c \cdot \text{Vol}(C).$$

This inequality indicates that the probability $\mu(C)$ of a newly sampled point falling within the convex hull $C$ increases with the volume $\text{Vol}(C)$. The compactness of $M$ ensures the validity of these properties, especially as the number of sampled points increases, leading to greater coverage in $M$.

