# OpenReview forum: "Understanding the Connection between Low-Dimensional Representation and Generalization via Interpolation"
_ICLR.cc/2025/Conference — ICLR 2025 Conference Withdrawn Submission_

### Official Review · Reviewer_Mv3t · 2024-11-04

**Soundness:** 1
**Presentation:** 1
**Contribution:** 1
**Rating:** 1
**Confidence:** 3

**Summary:**

This paper discusses how three concepts: dimension of representations, interpolation probability, and the volume of the data are connected. The findings of the paper are: lower dimension leads to tighter generalization bounds, higher dimension leads to a lower interpolation probability, lower data volume leads to a lower interpolation probabiliy. They verify the findings with experiments.

**Strengths:**

The authors try to verify their findings with experiments.

**Weaknesses:**

1. I didn't understand what interpolation regime is. Is it the regime that the training error is zero?

2. Also, there is no comparison with existing generalization bounds. It reads like the paper have proposed a new way of improving existing generalization bounds, but it is not. As their generalization bound is a naive concentration inequality, compared to state-of-the-art bounds I believe it would be weak, and would be hard to grasp the true phenomenon behind generalizatation.

3. Also, it is hard to see the detailed picture that the paper is trying to depict. What I can see is that this paper tries to connect d, the dimension of the representation, and the set of the representations in $\mathbb{R}^{d}$ to show that generalization and low-dimensionness is connected because it leads to "interpolating" rather than "extrapolating". Is the paper showing things that support the claim? I am not convinced. What they do is

- show that intrinsic dimension, interpolation probability, and convex hull volume changes over training.
- show that the dimension of the representation is related to generalization (here, it is not the intrinsic dimension but the dimension of the embedding)
- Larger dimension leads to smaller interpolation probability because it is harder to cover up the whole space when the dimension is large.
- Larger volume of the convex hull of sampled data leads to higher interpolation probability (which is quite straightforward from the definition of interpolation probability)

Two things I couldn't understand were:
-> Are generalization bounds and data volume directly related? for example, suppose data 1 has d = 1, volume = 100 and data 2 has d = 10, volume = 1000.  Surely data 2 will have larger interpolation probability. But the bound will be tighter when d=1.
-> Why are interpolation probabilities important? More specifically, where did you use the assumption of interpolation in Theorem 5.1?

4. Most of the claims in this paper are very straightforward or from different papers in the literature. The only new theorems are Thm 5.1 and Proposition 1, which are simply applying concentration inequalities or intergral inequalities. Hence this paper lacks theoretical novelty.

5. Why is letting $L = O(\sqrt{d})$ justifiable in Theorem 5.1? Actually $L = O(1/\sqrt{d})$, provided that the loss function is bounded in a fixed interval. That is because $L||x-y|| \approx ||f(x) - f(y)|| = O(1)$ and $||x|| = O(\sqrt{d})$.  With this scaling, $d$ does not depend on the generalization bound.

6. Some formatting errors: Proposition is labelled as 1, 2, ... whereas theorems are labelled as Thm 6.1, 6.2, .... In pg 7, you compare the interpolation probabilities for the triangle and circle twice.

**Questions:**

See Weaknesses.

---

### Official Review · Reviewer_WGn6 · 2024-11-04

**Soundness:** 2
**Presentation:** 3
**Contribution:** 3
**Rating:** 5
**Confidence:** 4

**Summary:**

The paper presents a theoretical framework that explores how low-dimensional representations in neural networks affect their generalization performance by examining interpolation and convex combinations. It posits that lower-dimensional embeddings increase the probability that new samples fall within the convex hull of the training data, thereby raising interpolation probability and reducing generalization error bounds. The authors support their framework with theoretical proofs and experiments on benchmark datasets, demonstrating that neural networks tend to learn lower-dimensional manifolds during training, which is associated with improved generalization. Additionally, the study investigates the influence of data manifold geometry, specifically convex hull volume, and training data size on interpolation probability. The results indicate that compact, low-dimensional embeddings contribute to better generalization performance, and the paper discusses the limitations of the proposed framework and suggests directions for future research.

**Strengths:**

The authors introduce a novel framework that connects low-dimensional representations in neural networks to their generalization performance through interpolation and convex combinations, a combination not extensively explored previously. The presentation is sound, featuring relevant theoretical contributions (a generalization error upper bound is proposed in the interpolation regime), complemented by empirical results using standard models to support the proposed claims. Clarity is maintained throughout the paper, with well-organized sections, clear definitions, and precise presentations of theorems and experimental results. The limitations are also clearly stated in discussion and conclusion.

**Weaknesses:**

The authors do not provide details regarding the considered neural network architectures and training hyperparameters, which hinders the reproducibility of the empirical results. The architectural details of the "5-layer MLP" in Section 4 of the main text are not provided, and there are no supplementary materials $-$ there is no code to reproduce the empirical results in the paper. Moreover, the experiments in the main text are confined to MNIST with a simple fully connected network, which exhibit the expected training dynamics of decreasing intrinsic dimension and increasing interpolation probability over time. However, as shown in the appendix, some complex architectures like AlexNet on CIFAR-10 do not follow this pattern, with the intrinsic dimension increasing after an initial decrease (whereas other architectures like VGG-16 do follow this pattern). This discrepancy raises concerns about the scalability of the proposed theoretical framework to higher-dimensional data and more sophisticated models. Additionally, the focus on interpolation may be overly constrained, as real-world applications often require models to generalize beyond the convex hull of training data, a scenario not adequately addressed by the theory (as the authors themselves have acknowledged). This oversight may limit the framework's relevance in practical settings where extrapolation is necessary for robust generalization.

**Questions:**

1. It would be helpful if the authors could provide the code and architectural details in the supplementary materials so that the results in the main text can be reproduced.

2. Have the authors compared the temporal behavior of the intrinsic dimension, convex hull volume, and interpolation probability on higher dimensional datasets (e.g., Imagenette or ImageNet)?

---

### Official Review · Reviewer_BSte · 2024-11-04

**Soundness:** 1
**Presentation:** 1
**Contribution:** 1
**Rating:** 3
**Confidence:** 3

**Summary:**

This paper studies generalization of machine learning models, and in particular neural networks. The paper claims to establish a connection between generalization and the structure of the data manifold. However, it is not clear to me what the actual contribution of this paper is besides a discussion

**Strengths:**

-

**Weaknesses:**

My main concern with this paper is the lack of novelty:
- Their main result, Theorem 5.1 is a very well known fact (see e.g., high dimensional statistics by Martin Wainwright)
- The proof of Theorem 5.1 uses McDiarmid, which requires that (x,y) are iid samples. The Theorem statement should include this assumption.
- It is not clear to me what the link is between Theorem 5.1 and the concept of ''Interpolation Probability''.
- Proposition 1 is trivial and should be merely stated as a fact or an observation.

**Questions:**

-

**Details Of Ethics Concerns:**

-

---

### Official Review · Reviewer_Q8tS · 2024-11-04

**Soundness:** 3
**Presentation:** 2
**Contribution:** 2
**Rating:** 3
**Confidence:** 3

**Summary:**

This work addresses the question of the connection between the low-dimensional representation and the generalization of neural networks both theoretically and empirically. The authors focus on the convex hull of training points and analyze the probability that a new point falls into this hull. The derived upper bound shows that such a probability sharply decreases for low Lipschitz constant, small diameter of the data points, and large ambient dimension. The numerical experiments justify this.

**Strengths:**

- This study presents a clear connection between the convex hull of training points and the generalization ability of neural networks.
- This study also provides empirical justification for their theoretical analysis.

**Weaknesses:**

I raise the following as the major weaknesses of this work.
1. Limited technical contributions
2. Poor paper writing

I elaborate on the weaknesses below.

1. Limited technical contributions
It has been widely known that low dimensional representation leads to generalization. This paper confirms this from the perspective of the convex hull of training points, which may be novel itself, but there is only one theoretical claim (Theorem 5.1). Even for this claim, the proof is relatively straightforward: an adaptation of McDiarmid's inequality. Proposition 1 is even more trivial. I acknowledge the results (both theoretical and empirical ones) but don't think they clear the bar of ICLR.

2. Poor paper writing
There is great room for improvement in the technical writing in this manuscript. Among others, there are many undefined symbols when the authors make technical claims. To list a few,
- [Eq. (4)] Definition of $\mathcal{R}(\,\cdot\,)$
- [Eq. (12)] Definition of $\Phi^n$
- [Eq. (12)] Definition of $i$
- [Eq. (13)] Definition of $d\lambda(x)$ and $\mathrm{Vol}(C)$.

**Questions:**

Please answer the two weaknesses raised above.

---

### Official Review · Reviewer_aPfK · 2024-11-04

**Soundness:** 2
**Presentation:** 3
**Contribution:** 1
**Rating:** 3
**Confidence:** 4

**Summary:**

This paper aims to understand the connection between generalization and low-dimensional representations. Specifically, it examines the convex hull of the network's features/outputs for the training data and is interested in the probability (interpolation probability) that a new data point lies in it. The paper experimentally shows that one measure of effective dimension decreases, while the interpolation probability increases during training.

**Reason for Score**

Overall the idea is ineresting but the paper does not do enough. I do not think the new theory results provide new information. Some of the experiments are interesting, but a lot of the take aways do not seem novel.

I think more thourough investigation into the concentration of outputs from neural networks (i.e. like nueral collapse, where the volume doesnt increase, but the interpolation probability does) could be interesting. Then connecting this to effective dimensionality could also be novel. However this would require many more experiments.

**Strengths:**

The paper is easy to follow, and the idea that at least some measure of effective dimensionality is reduced during training is interesting. I appreciate the papers attempt to unify generalization and low rankness.

**Weaknesses:**

The paper, I think, has a few issues.

1. The Lipschitzness of the loss function is too strong of an assumption and hides a lot of complexities. For example, let $f_\theta$ be my interpolating neural network. Let $x,y$ be any point in the training and $x',y'$ be any close points not in the training set. The Lipschitzness implies that the $x',y'$ loss is close to the loss for $x,y$ (which is 0), **regardless** of what $f(x')$ is, specifically, I can arbitrarily increase $\|f(x') -y'\|$ without changing the upper bound on the loss. Hence, this assumption is too strong, and the theorem does not say anything.

2. Similarly, proposition 2 doesn't say anything either. Like clearly, if the measure of a set increases, then the probability of being in the set should go up. Of course, they can be large measure low probability sets, but the result doesn't say anything about this.

3. Similarly, I am not sure what new insight is gained in Section 8. Yes, if we have more points then the convex has larger volume.

**Questions:**

In definition 3, what is $\mathcal{R}$, and how is both a set and a function (equation (4))?

---

### Note · Authors · 2024-11-23

I have read and agree with the venue's withdrawal policy on behalf of myself and my co-authors.